# Pulsed Electric Field Processing of Red Wine: Effect on Wine Quality and Microbial Inactivation

**Gulsun Akdemir Evrendilek**

Department of Food Engineering, Faculty of Engineering, Golkoy Campus, Bolu Abant İzzet Baysal University, 14030 Bolu, Turkey; gevrendilek@ibu.edu.tr; Tel.: +90-374-254-1000 (ext. 4858)

**Abstract:** Pulsed electric field (PEF) treatment of red wine samples with energies changing from 2.4 to 13.2 kJ to inactivate *Saccharomyces cerevisiae*, *Hansenula anomala*, *Candida lipolytica*, *Lactobacillus delbrueckii* ssp. *bulgaricus*, and *Escherichia coli* O157:H7 with the determination of the changes in the quality and sensory properties in addition to metal ion concentration (Na, Mg, K, and Mn) were explored. Increased applied energy resulted in a significant increase in pH, conductivity, lightness (*L\**), yellowness (*b\**), and total phenolic substance content with significant inactivation of all microorganisms with no significant change in metal ion concentration. Sensory properties of particle status, sour taste, and aftertaste were significantly decreased, whereas the other measured properties were significantly increased by 13.2 kJ PEF treatment ($p < 0.05$). Joint optimization studies for the most optimal processing parameters for the measured properties were 488 s, 0.13 kJ, and 0.22 kV; 488 s, 13.2 kJ, and 31 kV; 348 s, 9.39 kJ, and 31 kV/cm; and 488 s, 13.2 kJ, and 0 kV EFS, with 0.79, 0.69, 1.00, and 0.72 composite desirability, respectively.

**Keywords:** pulsed electric fields; red wine; microbial inactivation; wine quality

## 1. Introduction

With the great demand to eliminate and/or reduce antimicrobial agents, new processing technologies as an alternative to traditional ones have gained more attention. The wine industry, with no exception, is in search to increase the number of color compounds, phenolics, as well as anthocyanins in wine, and replace or reduce the amount of SO$_2$ used as an antioxidant and antimicrobial agent to inactivate molds in the early stages of wine, bacteria, and yeasts during fermentation; furthermore, preventing microbial spoilage during wine production process and storage, and termination of fermentation with the elimination a fermentative yeast, *Saccharomyces cerevisiae*, due to its adverse effect on human health such as asthma, allergy, and headache [1]. Thus, it is of great interest to the wine industry to find alternatives to provide healthy, safe, and high-quality products with low concentrations of chemical preservatives [2].

Novel processing technologies can bring new alternatives to the wine industry with an improvement of competitiveness by introducing new products, upgrading food quality, and reducing energy costs [3]. Pulsed electric fields (PEF) is one of the most promising nonthermal novel processing technologies that is utilized to increase the extraction of anthocyanins, enhancement of the main pigments responsible for the color of the grapes [4,5], total phenolic content, the anthocyanin concentration [6], polyphenols from skins of Chardonnay, a white grape variety [7], improvement of chromatic parameters, and the phenolic content of Cabernet Sauvignon red wines during wine-making [8]. Moreover, it is used to control spoilage microbiota during vinification with the inactivation of *Dekkera bruxellensis*, *Dekkera anomala*, *Lactobacillus hilgardii*, and *Lactobacillus plantarum* in must and wine [9] and to inactivate *Escherichia coli* O157:H7, *Candida lipolytica, Hansenula anomala*, *Lactobacillus bulgaricus*, and *Saccharomyces cerevisiae* in red wine [10].

Studies conducted with PEF are mostly focused on the processing of must and pomace with efficacy in the extraction of bioactive compounds, inactivation of wine spoilage

microflora, and acceleration of wine aging [11–13]; however, PEF treatment of red wine samples with the determination of changes in physicochemical and sensory properties in addition to microbial inactivation are very limited. Thus, the objectives of the current research are to process young red wine samples for the termination of fermentation by means of PEF to reduce $SO_2$ use and to determine changes in physicochemical and sensory properties with the inactivation of microbiota.

## 2. Materials and Methods

### 2.1. Red Wine Samples

The wine samples from Okuzgozu grapes (Elazig Province of Turkey) with 12–13% alcohol content before the completion of fermentation were kindly provided by Dimes Gida San ve Tic A.S. (Tokat, Turkey). The samples, after receiving, were processed by PEF immediately; both control and PEF-treated samples at room temperature were subjected to analyses.

### 2.2. Test Microorganisms and Their Inoculation into Wine Samples

The cultures of *Saccharomyces cerevisiae*, *Hansenula anomala*, *Candida lipolytica* (*Yarrowia lipolytica*), and *Lactobacillus delbrueckii* ssp. *bulgaricus* were obtained from the Ankara University culture collection (Ankara, Turkey).

The yeasts were activated by transferring them from tryptic soy agar (TSA, Fluka, Munich, Germany) slants into tryptic soy broth (TSB) (Fluka) following incubation at $22 \pm 2$ °C for 12 h. After that, they were inoculated in wine samples separately. *L. delbrueckii* ssp. *bulgaricus* culture was inoculated into wine samples after transferring it to MRS broth (Fluka) and following incubation at $35 \pm 2$ °C for 12 h. *Escherichia coli* O157:H7 (EDL 931 04054) culture from the Refik Saydam Hıfzıssıhha Research Center Culture Collection Laboratory (Ankara, Turkey) in lyophilized form was activated by transferring into McConkey Sorbitol Agar (Fluka) following incubation at $35 \pm 2$ °C. Activated cultures were inoculated into wine samples, separately.

### 2.3. PEF Treatment

The bench scale PEF treatment unit (OSU-4A, Ohio State University, Columbus, OH, USA) equipped with six treatment chambers having a 0.29 cm diameter and 0.23 cm gap was used to treat wine samples. The system provided square wave bipolar pulses with a 20 µs pulse delay time. K-type dual channel digital thermocouples (Fisher Scientific, Pittsburgh, PA, USA) were placed at the inlet and the outlet of each pair of treatment chambers to monitor the pre- and post-treatment temperatures (T2–T1, T4–T3, and T6–T5) of 14–12, 14–13, 14–13 °C, respectively. In order to control the temperature of the samples during PEF treatment, the treated sample was cooled after each pair of chambers by cooling coils submerged in a water bath (model RTE-111; NESLAB Instruments, Inc., Newington, NH, USA) at 10–12 °C. A PEF processing unit was donated with a trigger generator (model 9300 series; Quantum Composers, Inc., Bozeman, MT, USA) to control pulse delaying time, pulse duration time, and pulse repetition rate. A two-channel digital oscilloscope (model TDS 320; Tektronix, Inc., Beaverton, OR, USA) was utilized to measure applied voltage and current. OSU-4A bench scale PEF units had 60 A max of output current, 200–1200 Ω of load resistance, 12,000 V max of output voltage, 16 J of energy storage, and 10,000 pulse per second (pps) max of repetition rate when fully charged.

Preliminary tests were conducted to determine PEF treatment parameters to process red wine samples. Three µs of pulse duration, 20 µs of pulse delay time, 40 mL/min of flow rate, and 500 pps of frequency with 0, 17, 24, 31 kV/cm electric field strengths, 0, 163, 325, and 488 µs treatment times and 0, 2.4, 3.4, 4.4, 4.9, 6.8, 7.3, 8.8, 10.2, and 13.2 kJ energies were utilized.

### 2.4. Red Wine Physicochemical Analyses

pH was determined on 10 mL samples at room temperature by an Orion perpHect logR meter (inoLab WTW, Weilheim, Germany). Titratable acidity (TA) measurement was conducted with reference to tartaric acid (g/L), whereas conductivity (mS/cm) measurement of the samples was performed by a handheld conductivity meter (Sension 5 model, HACH, Loveland, CO, USA). Color parameters of lightness ($L^*$), redness ($a^*$), and yellowness ($b^*$) were measured by a Hunter color flex spectrophotometer (Hunter Associates Laboratory, Inc., Reston, VA, USA) using the CIELAB color scale at D65/10°. Hue and chroma values of the wine samples were calculated from the lightness ($L^*$), $a^*$, and $b^*$ values. All measurements were conducted at room temperature (22 ± 2 °C).

Total monomeric anthocyanin (TMAC) content was determined on the samples diluted with 0.025 M KCl (Sigma Chemical, Co., Stockholm, Sweden) and 0.04 M sodium acetate (Sigma Chemical, Co., Stockholm, Sweden), separately. The dilutions were centrifuged at 2400 rpm for 2 min and the absorbances were read at both 520 and 700 nm (Lambda 25 model spectrophotometer, Perkin Elmer, Waltham, MA, USA) after setting the samples for 20 min. Results were reported as cyanidin 3-glucoside equivalent in mg/L [10].

The samples were mixed with a Tris-HCl tampon at pH 7.4 and the mixture was vortexed at 2400 rpm for 5 min before the addition of 1 mL of 1,1-diphenyl-2-picrylhydrazyl (DPPH) prepared in ethanol. An absorbance measurement was performed at 517 nm for the determination of the total antioxidant activity (% TAC) [14].

Total phenolic substance content (TPSC) as mg GAE/L of the samples was determined after the diluted samples were filtered through the 0.45 μm filter before mixing with 0.2 N Folin-Ciocalteu and sodium carbonate. The mixture waited in a water bath at 50 ± 5 °C for 5 min, and the absorbance measurement at 760 mn was performed after cooling down to room temperature. A calibration curve was prepared at 100, 200, 300, 400, and 500 mg/L gallic acid solution [15].

Metal ion concentration with analytical masses of 75As, 40Ca, 24Mg, 111Cd, 63Cu, 52Cr, 56Fe, 202Hg, 39K, 23Na, 24Mg, 55Mn, 31P, 208Pb, 80Se, 118Sn, and 66Zn were determined by inductively coupled plasma mass spectroscopy (ICP-MS) (XSERIES 2, Thermo Scientific, Schwerte, Germany). A calibration curve with concentrations from 2.5 to 1000 mg/kg was prepared for each element. Limits of detection (LOD) and limit of quantification (LOQ) values were utilized, and recovery values were reported to be 70–120% [15].

### 2.5. Microbiological Analyses

After dilution of the samples with 0.1% peptone (Fluka) water, appropriate dilutions for *E. coli* O157:H7 were plated onto MacConkey Sorbitol Agar (Fluka) plates, dilutions for *H. anomala*, *C. lipolytica*, and *S. cerevisiae* were plated onto Yeast Extract Peptone Glucose Agar (Fluka), and dilutions for *L. delbrueckii* ssp. *bulgaricus* were plated onto MRS agar (Fluka). *E. coli* O157:H7 and *L. delbrueckii* ssp. *bulgaricus* plates were incubated at 35 ± 2 °C for 24–48 h under anaerobic conditions, whereas *H. anomala*, *C. lipolytica*, and *S. cerevisiae* plates were incubated at 22 ± 2 °C for 3–5 days under aerobic conditions.

### 2.6. Sensory Analyses

The samples at room temperature were served to 15 trained panelists for three-step sensory analyses of visual evaluation, smell, and taste. The panelists were asked to evaluate the samples for cloudiness/clearness, dullness/brightness, red color intensity, density, and particle status by tilting and holding the glass in front of the given white paper for visual analyses. They were asked to swirl the glasses for 0–12 s, take a quick whiff to gain a first impression, and take a breath for receiving the whole impression of the smell. They were also asked to taste the samples by taking a sip, rolling it all around the mouth, and evaluating the samples in three stages; the initial attack, the evaluation, and at the end of the evaluation. At the end, they were instructed to swallow the samples for the evaluation of aftertaste. During the evaluation of the samples, they drank water, consumed a couple

bits of unsalted cracker, and drank water again. Sensory analyses were performed based on a 9-point hedonic scale [10].

### 2.7. Data Analyses

Data were analyzed by one-way analysis of variance (ANOVA) for the evaluation of the significance (F test) and Tukey's multiple comparison test was utilized to determine the means significant differences in quality properties (pH, TA, conductivity, lightness (*L\**), *a\**, *b\**, TPSC, TAC, and TMAC), microbial inactivation (*C. lipoliytica*, *S. cerevisiae*, *H. anemola*, *L. delbrueckii* ssp. *bulgaricus*, and *E. coli* O157:H7), changes in metal ion concentration (Na, Mg, K, and Mn), and the sensory properties (cloudiness/clearness, dullness/brightness, red color intensity, density, particle status, odor/flavor, bitterness, sour taste, and after taste) by Minitab 17 (Minitab, Inc., State College, PA). The best-fit multiple (non-)linear regression (MNLR) models for 24 response variables in addition to joint optimization were also built by taking into account the composite desirability function (D), a geometric average of individual desirability. The joint optimization involved the minimization of the responses for the microbial loads and the target values for the physicochemical properties and sensory properties. Each experiment was repeated three times.

### 3. Results and Discussions

Processing of red wine by PEF up to 13.2 kJ caused a significant increase in pH, conductivity, lightness (*L\**), *b\**, and TPSC ($p < 0.05$). Even though an increase was observed for TA, *a\**, TAC, and TMAC, the observed difference was not significant ($p \leq 0.05$) (Table 1). The mean initial count of *C. lipoliytica*, *S. cerevisiae*, and *H. anemola* yeasts, $6.25 \pm 0.76$, $7.868 \pm 1.13$, and $6.30 \pm 1.34$, reduced to $0.60 \pm 0.07$, $0.50 \pm 0.16$, and $0.33 \pm 0.12$ revealing 5.65, 7.86, and 5.97 log inactivation, respectively. The mean initial *L. delbrueckii* ssp. *bulgaricus* and *E. coli* O157:H7 counts, $6.11 \pm 0.23$ and $6.40 \pm 0.55$, reduced to $1.09 \pm 0.36$ and $0.49 \pm 0.14$, resulting in 5.02 and 5.01 log reductions, consequently (Table 2).

**Table 1.** Changes in the physicochemical properties of red wine samples treated by pulsed electric fields.

| Energy (kJ) | pH | TA (g of Tartaric Acid/L) | Conductivity (mS/cm) | *L\** | *a\** | *b\** | TPSC (mg GAE/L) | TAC (%) | TMAC (mg/L) |
|---|---|---|---|---|---|---|---|---|---|
| 0.0 | 2.94 ± 0.04 c | 1.98 ± 0.04 a | 2.27 ± 0.02 d | 13.92 ± 0.60 ab | 39.85 ± 2.17 abc | 22.21 ± 0.95 f | 2212.5 ± 23.4 c | 72.68 ± 0.45 b | 37.19 ± 0.46 e |
| 2.4 | 2.96 ± 0.04 abc | 1.20 ± 0.01 a | 2.27 ± 0.01 cd | 12.03 ± 0.64 cd | 38.73 ± 0.05 abcd | 21.10 ± 0.07 f | 2225 ± 18.5 bc | 72.65 ± 0.08 b | 37.09 ±0.15 de |
| 3.4 | 2.97 ± 0.03 abc | 2.00 ± 0.00 a | 2.26 ± 0.02 d | 10.66 ± 0.56 d | 34.47 ± 0.10 d | 21.43 ± 0.06 e | 2339 ± 35.2 abc | 72.83 ± 0.63 ab | 37.12 ± 0.04 de |
| 4.4 | 3.00 ±0.01 abc | 1.98 ± 0.01 a | 2.30 ± 0.01 bcd | 10.82 ± 0.59 d | 37.93 ± 0.05 bcd | 21.64 ± 0.06 d | 2424 ± 1.82 ab | 72.86 ± 0.63 ab | 38.01 ± 0.23 bcd |
| 4.9 | 2.93 ± 0.01 bc | 1.99 ± 0.01 a | 2.30 ± 0.01 bcd | 13.13 ± 0.03 abc | 39.70 ± 0.06 abc | 21.80 ± 0.07 c | 2295 ± 6.26 abc | 73.34 ± 0.77 ab | 37.67 ± 0.24 cde |
| 6.8 | 2.96 ± 0.02 abc | 2.00 ± 0.00 a | 2.31 ± 0.01 abc | 13.58 ± 1.50 abc | 41.04 ± 0.09 a | 22.43 ± 0.06 b | 2366.3 ± 38.5 abc | 73.18 ± 0.86 ab | 37.51 ± 0.43 cde |
| 7.3 | 2.98 ± 0.01 abc | 1.99 ± 0.01 a | 2.31 ± 0.01 abc | 14.76 ± 0.05 a | 41.85 ± 0.09 ab | 23.94 ± 0.17 a | 2382 ± 6.44 abc | 73.58 ± 0.23 ab | 37.99 ± 0.08 bcd |
| 8.8 | 3.03 ± 0.01 ab | 1.99 ± 0.01 a | 2.32 ± 0.02 ab | 12.01 ± 0.96 cd | 37.08 ± 1.05 cd | 17.64 ± 0.06 c | 2477 ± 6.85 a | 73.49 ± 0.35 ab | 38.70 ± 0.32 ab |
| 10.2 | 3.03 ± 0.06 b | 1.20 ± 0.01 a | 2.33 ± 0.01 ab | 12.71 ± 0.34 bc | 36.83 ± 0.53 cd | 14.82 ± 0.62 d | 2373.3 ± 29.1 abc | 73.58 ± 0.42 ab | 38.24 ± 0.12 abc |
| 13.2 | 3.05 ± 0.02 a | 1.97 ± 0.01 a | 2.34 ± 0.01 a | 13.06 ± 0.09 abc | 40.22 ± 0.05 abc | 21.20 ± 0.10 b | 2494. 59 ± 2.12 a | 74.04 ± 0.18 a | 39.25 ± 0.37 a |

Data in the same column with different superscript letters are significantly different ($p < 0.05$). TA: Total acidity (g of tartaric acid/L). *L\** (Lightness), *a\** (redness), *b\** (yellowness). TPSC: Total phenolic substance content (mg GAE/L). TAC: Total antioxidant capacity (%); TMAC: Total monomeric anthocyanin content (mg/L).

**Table 2.** Inactivation of yeasts and bacteria in red wine samples treated by pulsed electric fields.

| Energy (kJ) | *C. lipoliytica* | *S. cerevisiae* | *H. anemola* | *L. delbrueckii* ssp. *bulgaricus* | *E. coli* O157:H7 |
|---|---|---|---|---|---|
| 0.0 | 6.25 ± 0.76 a | 7.86 ± 1.12 a | 6.30 ± 1.34 a | 6.11 ± 0.23 a | 6.40 ± 0.55 a |
| 2.4 | 3.54 ± 0.11 b | 5.13 ± 0.11 b | 5.03 ± 0.02 ab | 5.52 ± 0.16 b | 4.70 ± 0.33 b |
| 3.4 | 3.17 ± 0.27 b | 3.96 ± 0.47 bc | 4.77 ± 0.09 b | 5.30 ± 0.26 b | 4.67 ± 0.12 b |
| 4.4 | 3.02 ± 0.14 b | 3.41 ± 0.24 bcd | 4.34 ± 0.32 b | 5.19 ± 0.03 b | 3.48 ± 0.35 c |
| 4.9 | 2.85 ± 0.14 b | 3.16 ±± 0.53 bcde | 4.05 ± 0.20 bc | 4.29 ± 0.23 c | 3.23 ± 0.08 abcd |
| 6.8 | 2.39 ± 0.51 bc | 2.63 ± 0.52 cdef | 3.74 ± 0.08 bcd | 4.04 ± 0.02 c | 2.15 ± 0.36 cd |
| 7.3 | 1.05 ± 0.08 cd | 2.43 ± 0.09 cdef | 3.08 ± 0.12 bcd | 3.30 ± 0.33 d | 1.48 ± 0.09 ef |
| 8.8 | 1.02 ± 0.03 cd | 1.63 ± 0.42 def | 1.96 ± 0.07 cde | 3.00 ± 0.08 d | 1.20 ± 0.18 ef |
| 10.2 | 0.95 ± 0.07 cd | 0.89 ± 0.26 ef | 1.64 ± 0.06 de | 2.28 ± 0.34 e | 1.01 ± 0.17 ef |
| 13.2 | 0.60 ± 0.07 d | 0.50 ± 0.16 f | 0.33 ± 0.12 e | 1.09 ± 0.36 f | 0.49 ± 0.14 f |

Data in the same column with a different superscript letter are significantly different ($p < 0.05$).

The mean initial metal ion concentration of the red wine samples; Na with $80.00 \pm 5.77$ µg/kg, Mg with $339.62 \pm 26.43$ µg/kg, K with $2246.20 \pm 136.00$ µg/kg, and Mn with $10.36 \pm 3.67$ µg/kg did not significantly change by the applied PEF treatments. Ca, Cd, Cu, Cr, Fe, Hg, P, Pb, Se, Sn, and Zn of the samples were not determined as they were below the detection limit (Table 3).

**Table 3.** Changes in the metal ion concentration of the red wine samples treated by pulsed electric fields.

| Energy (kJ) | Na (µg/mL) | Mg (µg/mL) | K (µg/mL) | Mn (µg/mL) |
|---|---|---|---|---|
| 0.0 | $80.00 \pm 5.77$ [a] | $339.62 \pm 26.43$ [a] | $2246.20 \pm 136.00$ [a] | $103.6 \pm 3.67$ [a] |
| 2.4 | $73.30 \pm 4.46$ [a] | $310.00 \pm 31.40$ [a] | $2301.11 \pm 264.08$ [a] | $11.67 \pm 3.87$ [a] |
| 3.4 | $76.56 \pm 2.28$ [a] | $350.40 \pm 39.30$ [a] | $2224.54 \pm 186.89$ [a] | $7.18 \pm 2.55$ [a] |
| 4.4 | $77.30 \pm 2.93$ [a] | $366.73 \pm 12.34$ [a] | $2121.70 \pm 120.10$ [a] | $6.82 \pm 0.82$ [a] |
| 4.9 | $72.51 \pm 5.45$ [a] | $362.20 \pm 20.20$ [a] | $2134.00 \pm 152.50$ [a] | $9.92 \pm 1.19$ [a] |
| 6.8 | $75.78 \pm 3.00$ [a] | $350.65 \pm 28.10$ [a] | $2303.90 \pm 33.40$ [a] | $7.35 \pm 0.28$ [a] |
| 7.3 | $75.75 \pm 6.07$ [a] | $347.30 \pm 30.30$ [a] | $2366.00 \pm 103.33$ [a] | $8.89 \pm 1.44$ [a] |
| 8.8 | $77.38 \pm 2.76$ [a] | $340.80 \pm 54.32$ [a] | $2307.70 \pm 105.00$ [a] | $6.78 \pm 0.38$ [a] |
| 10.2 | $73.75 \pm 6.07$ [a] | $351.65 \pm 28.98$ [a] | $2266.50 \pm 155.80$ [a] | $7.13 \pm 0.68$ [a] |
| 13.2 | $77.38 \pm 5.55$ [a] | $350.29 \pm 39.58$ [a] | $2273.60 \pm 72.50$ [a] | $6.64 \pm 0.24$ [a] |

Data in the same column with a different superscript letter are significantly different ($p < 0.05$).

Among the mean initial sensory properties of cloudiness/clearness ($8.19 \pm 0.93$), dullness/brightness ($7.67 \pm 1.11$), red color intensity ($8.00 \pm 0.89$), density ($7.48 \pm 0.68$), particle status ($2.48 \pm 0.68$), odor/flavor ($7.62 \pm 0.74$), bitterness ($1.91 \pm 0.62$), sour taste ($3.04 \pm 0.74$), and after taste ($4.09 \pm 0.80$), only particle status ($1.00 \pm 0.00$), sour taste ($1.00 \pm 0.00$), and after taste ($2.24 \pm 0.36$) were significantly decreased by the 13.2 kJ PEF treatment. PEF-treated samples were evaluated as having a sweeter taste, less sour aftertaste, and fewer or smaller particles being clearer. Although higher red color intensity, with more brightness and clear color was reported, these differences were not significant (Table 4).

**Table 4.** Changes in the sensory properties of red wine samples treated by pulsed electric fields.

| Energy (kJ) | Cloudiness/ Clearness | Dullness/ Brightness | Red Color Intensity | Density | Particle Status | Odor/Flavor | Bitterness | Sour Taste | After Taste |
|---|---|---|---|---|---|---|---|---|---|
| 0.0 | $8.19 \pm 0.93$ [a] | $7.67 \pm 1.11$ [a] | $8.00 \pm 0.89$ [a] | $7.48 \pm 0.68$ [a] | $2.48 \pm 0.68$ [a] | $7.62 \pm 0.74$ [a] | $1.91 \pm 0.62$ [a] | $3.04 \pm 0.74$ [a] | $4.09 \pm 0.80$ [a] |
| 2.4 | $8.67 \pm 0.52$ [a] | $7.00 \pm 1.00$ [a] | $8.00 \pm 1.00$ [a] | $7.34 \pm 0.58$ [a] | $3.00 \pm 1.00$ [a] | $7.67 \pm 0.58$ [a] | $2.67 \pm 0.58$ [a] | $3.34 \pm 0.58$ [ab] | $4.00 \pm 1.00$ [a] |
| 3.4 | $7.00 \pm 0.00$ [a] | $7.00 \pm 2.00$ [a] | $7.68 \pm 0.60$ [a] | $8.34 \pm 0.60$ [a] | $1.68 \pm 0.58$ [ab] | $7.60 \pm 1.16$ [a] | $2.33 \pm 0.52$ [a] | $2.67 \pm 0.58$ [abc] | $3.68 \pm 0.58$ [a] |
| 4.4 | $7.33 \pm 0.58$ [a] | $7.33 \pm 0.58$ [a] | $8.00 \pm 1.00$ [a] | $7.70 \pm 0.80$ [a] | $1.68 \pm 0.60$ [ab] | $7.33 \pm 0.58$ [a] | $1.68 \pm 0.52$ [a] | $2.00 \pm 0.00$ [a] | $3.72 \pm 0.68$ [a] |
| 4.9 | $7.67 \pm 0.88$ [a] | $8.00 \pm 1.00$ [a] | $7.60 \pm 1.00$ [a] | $7.60 \pm 0.60$ [a] | $2.68 \pm 0.60$ [ab] | $7.00 \pm 1.00$ [a] | $2.34 \pm 0.60$ [a] | $3.34 \pm 0.58$ [ab] | $4.00 \pm 1.00$ [a] |
| 6.8 | $7.00 \pm 0.00$ [a] | $7.68 \pm 0.60$ [a] | $8.00 \pm 1.00$ [a] | $7.40 \pm 0.80$ [a] | $2.34 \pm 0.70$ [ab] | $7.70 \pm 1.53$ a | $1.34 \pm 0.58$ [a] | $2.34 \pm 0.52$ [abc] | $3.34 \pm 0.62$ [a] |
| 7.3 | $8.00 \pm 1.00$ [a] | $8.00 \pm 1.00$ [a] | $8.60 \pm 0.70$ [a] | $8.30 \pm 0.70$ [a] | $2.24 \pm 0.60$ [ab] | $7.34 \pm 0.58$ [a] | $2.68 \pm 0.58$ [a] | $3.00 \pm 0.00$ [ab] | $4.00 \pm 1.20$ [a] |
| 8.8 | $7.00 \pm 0.00$ [a] | $7.33 \pm 0.60$ [a] | $7.70 \pm 0.80$ [a] | $7.40 \pm 0.60$ [a] | $1.70 \pm 0.70$ [ab] | $7.20 \pm 0.54$ [a] | $2.34 \pm 0.50$ [a] | $1.68 \pm 0.58$ [bc] | $3.68 \pm 0.48$ [a] |
| 10.2 | $7.00 \pm 1.00$ [a] | $7.80 \pm 0.80$ [a] | $8.33 \pm 0.60$ [a] | $8.00 \pm 1.00$ [a] | $2.70 \pm 0.56$ [ab] | $7.00 \pm 1.00$ [a] | $2.34 \pm 0.50$ [a] | $2.68 \pm 0.60$ [abc] | $3.66 \pm 0.48$ [ab] |
| 13.2 | $7.67 \pm 0.58$ [a] | $7.60 \pm 0.60$ [a] | $9.70 \pm 0.70$ [a] | $8.34 \pm 0.58$ [a] | $1.00 \pm 0.00$ [b] | $7.34 \pm 0.60$ [a] | $2.00 \pm 0.00$ [a] | $1.00 \pm 0.00$ [c] | $3.62 \pm 0.54$ [a] |

Data in the same column with a different superscript letter are significantly different ($p < 0.05$).

The regression model and the equation along with $R^2$, $R^2_{adj}$, and $R^2_{pred}$ values for each response parameter and microbial inactivation are depicted in Table 5. The highest $R^2$ and $R^2_{adj}$ values for the physicochemical properties of TPSC with 95.50 and 95.08%; pH with 92.78 and 91.95%, and TMAC with 88.43 and 86.73% were obtained. The highest $R^2$ and $R^2_{adj}$ values for the microbial inactivation of *C. lipoliytica* with 92.99 and 91.33%, *S. cerevisiae* with 95.11 and 93.95%, *H. anemola* with 96.30 and 95.43%, *L. delbrueckii* ssp. *bulgaricus* with 98.41 and 98.03%, and *E. coli* O157:H7 with 97.06 and 96.37% were obtained from the model, respectively (Table 5).

**Table 5.** Regression analyses of the physicochemical properties and microbial inactivation of red wine samples treated by pulsed electric fields.

| Measured Properties | Regression Equation | $R^2$ (%) | $R^2_{adj}$ (%) | $R^2_{pred}$ (%) | SE |
|---|---|---|---|---|---|
| pH | $=-0.000888E + 0.1174EFS + 0.01962Trt - 0.000028Trt*Trt + 0.000001E*Trt$ | 92.78 | 91.95 | 92.43 | 0.84 |
| Conductivity (mS/cm) | $=2.25885 - 0.000003E + 0.000202EFS + 0.000107Trt + 0.000000E*EFS$ | 88.65 | 87.59 | 86.84 | 0.009 |
| TA (g of tartaric acid/L) | $=1.9785 - 0.000615EFS + 0.000197Trt - 0.000000Trt*Trt$ | 20.12 | 14.68 | 2.70 | 0.27 |
| Lightness (*L\**) | $=14.025 - 0.0493EFS$ | 19.43 | 17.67 | 13.69 | 1.81 |
| *b\** | $=22.267 - 0.002033E - 0.00117Trt + 0.000004E*Trt$ | 54.44 | 51.34 | 45.65 | 2.38 |
| TPSC (mg GAE/L) | $=2017.6 + 4.22EFS + 0.6271Trt + 0.2584EFS*EFS - 0.01372EFS*Tr$ | 95.50 | 95.08 | 94.51 | 28.90 |
| TAC (%) | $=2.862EFS + 0.4908Trt - 0.000714Trt*Trt - 0.01930EFS*Trt + 0.000028EFS*Trt*Trt$ | 58.15 | 56.29 | 53.26 | 0.41 |
| TMAC (mg/L) | $=36.733 + 0.02469E - 0.0326EFS + 0.002216Trt + 0.001734EFS*EFS + 0.000025E*EFS - 0.02213EFS*Trt$ | 88.43 | 86.73 | 83.55 | 0.268 |
| *C. lipolytica* inactivation | $=6.366 - 0.00521Trt - 0.0188E - 0.1437EFS - 0.000001Trt*Trt + 0.000000E*E + 0.00149EFS*EFS - 0.000001Trt*E + 0.0172Trt*EFS - 0.000030E*EFS$ | 92.99 | 91.33 | 88.64 | 0.49 |
| *S. cerevisiae* inactivation | $=8.375 - 0.00452Trt + 0.0224E - 0.0106EFS - 0.000016Trt*Trt - 0.000000E*E - 0.00205EFS*EFS + 0.000003Trt*E - 0.0212Trt*EFS + 0.000048E*EFS$ | 95.11 | 93.95 | 91.86 | 0.61 |
| *H. anemola* inactivation | $=7.005 - 0.00637Trt + 0.0003E - 0.0511EFS - 0.000004Trt*Trt - 0.000000E*E - 0.00022EFS*EFS + 0.000000Trt*E - 0.0002Trt*EFS - 0.000004E*EFS$ | 96.30 | 95.43 | 93.38 | 0.37 |
| *L. delbrueckii* ssp. *bulgaricus* inactivation | $=6.151 - 0.00275Trt - 0.01141E + 0.0072EFS + 0.000006Trt*Trt + 0.000000E*E - 0.000087EFS*EFS - 0.000001Trt*E + 0.01016Trt*EFS - 0.000013E*EFS$ | 98.41 | 98.03 | 97.32 | 0.22 |
| *E. coli* O157:H7 inactivaton | $=6.456 - 0.00261Trt - 0.0018E + 0.0149EFS - 0.000012Trt*Trt + 0.000000E*E - 0.00145EFS*EFS + 0.000000Trt*E + 0.0013Trt*EFS - 0.000009E*EFS$ | 97.06 | 96.37 | 95.54 | 0.38 |

$R^2$ (%): Coefficient of determination. $R^2_{adj}$ (%): Corrected goodness-of-fit. $R^2_{pred}$: Predicted coefficient of determination. SE: Standard error.

The pH of the samples was significantly affected by EFS, Trt, E, the square term of Trt*Trt, and the interaction of E*Trt; conductivity by E, EFS, Trt, and the interaction of E*EFS; TA by Trt, the square term of Trt*Trt, and the interaction of EFS*Trt*Trt; the lightness (*L\**) value by EFS and the interactions of E*Trt and EFS*Trt; the *b\** value by E and the interaction of E*Trt (Table S1); TAC by EFS and Trt; TPSC by EFS, Trt, the square term of EFS*EFS, and the interaction of EFS*Trt, and TMAC by E and Trt, and the interactions of E*EFS and EFS*Trt, respectively (Table S2). Inactivation of *C. lipolytica* was significantly affected by EFS and the square term of Trt*Trt; *S. cerevisiae* by the square term of Trt*Trt and the interactions of EFS*Trt and E*Trt; *H. anemola* by Trt; *E. coli* O157:H7 by the interaction of Trt*Trt, and *L. delbrueckii* ssp. *bulgaricus* by Trt and the square term of Trt*Trt, respectively (Table S3).

The three best joint optimizations for red wine samples were presented in Table 6. The most optimal processing parameters for the quality properties were 488 s treatment time, 0.13 kJ energy, and 0.22 kV EFS with a D value of 0.79. The joint optimization with the most optimal treatment for microbial inactivation was 488 s treatment time, 13.2 kJ energy, and 31 kV EFS with a D value of 0.69. While the sensory properties of the PEF-treated red wine samples had the most optimal conditions of 348 s treatment time, 9.39 kJ energy, and 31 kV/cm EFS with 1.00 desirability, these conditions for metal ion concentrations were 488 s treatment time, 13.2 kJ energy, and 0 kV EFS.

Due to its ability to inactivate microorganisms and induction of the membrane permeabilization of grape skin cells with the application of low energy and short processing time, PEF is mostly used to improve wine production with a high content of phenolic compounds with better sensory qualities and stable color [9,12].

PEF treatment of red wine with a 31 kV/cm electric field strength did not cause significant changes in pH, conductivity, TA, lightness (*L\**), *a\**, and *b\**. Moreover, TAC, TPSC, and TMAC, even though there were fluctuations, were not significantly changed by applied electric field strength [10]. A 1% increase in pH of red wine inoculated with *Brettanomyces bruxellensis* and a 4% increase in red wine inoculated with *B. bruxellensis* and *Oenococcus oeni* 9304 after PEF treatment with no change in the total acidity of the *Pseudomonas parvulus* wines or a decrease in the other wines was reported after PEF treatment. A decrease in lactic acid concentration, but an increase in tartaric acid with the exception of *O. oeni* 0608 inoculated wines was also reported after PEF treatment [16]. An increase in pH can be observed as a consequence of several reactions including metal ion migration from the electrodes, growth of the yeasts as they have the ability to degrade organic acids [17], and the leakage of intercellular components from cells to liquid phase as a result of increased

membrane permeability. In parallel to the increase in pH, the conductivity of the samples was also significantly increased by the applied magnitude of energy.

**Table 6.** Three joint optimizations of the responses variables (R) as a function of the applied energies in the range of 2.4 and 13.2 kJ for red wine treatment.

| Responses | Solution 1 | Solution 2 | Solution 3 |
|---|---|---|---|
| pH | 2.95 | 2.95 | 2.95 |
| *L** | 14.02 | 14.02 | 14.02 |
| *a** | 40.14 | 40.14 | 40.14 |
| *b** | 21.67 | 19.45 | 21.69 |
| TAC (%) | 73.32 | 73.32 | 73.32 |
| TPSC (mg GAE/L) | 2332.62 | 2332.62 | 2332.62 |
| TMAC (mg/L) | 41.10 | 36.31 | 37.81 |
| Composite desirability | 0.79 | 0.75 | 0.70 |
| Optimal value | | | |
| Trt (s) | 488 | 488 | 488 |
| E (kJ) | 0.13 | 0.32 | 3.88 |
| EFS (kV) | 0.22 | 1.46 | 2.34 |
| *H. anemola* ($\log_{10}$ unit) | 2.511 | 3.442 | 4.511 |
| *S. cerevisiae* ($\log_{10}$ unit) | 2.342 | 3.442 | 4.511 |
| *C. lipolytica* ($\log_{10}$ unit) | 2.118 | 2.692 | 3.271 |
| *L. delbrueckii* ssp. *bulgaricus* ($\log_{10}$ unit) | 3.371 | 3.964 | 4.932 |
| *E. coli* O157:H7 ($\log_{10}$ unit) | 2.189 | 3.307 | 3.739 |
| Composite desirability | 0.69 | 0.65 | 0.62 |
| Optimal value | | | |
| Trt (s) | 488 | 488 | 488 |
| E (kJ) | 13.2 | 10.2 | 8.80 |
| EFS (kV) | 32 | 24 | 17 |
| Density | 6.19 | 6.77 | 6.77 |
| Red color intensity | 8.71 | 8.86 | 8.86 |
| Dullness/brightness | 7.09 | 6.98 | 6.98 |
| Cloudiness/clearness | 6.55 | 7.60 | 7.60 |
| Bitterness | 1.28 | 1.00 | 1.00 |
| Sour taste | 1.14 | 2.71 | 2.71 |
| Odor/flavor | 7.92 | 7.93 | 7.93 |
| Aftertaste | 2.44 | 3.62 | 3.62 |
| Composite desirability | 0.72 | 0.64 | 0.64 |
| Optimal value | | | |
| Trt (s) | 348 | 488 | 488 |
| E (kJ) | 9.39 | 2.04 | 2.04 |
| EFS (kV) | 31 | 2.31 | 2.31 |
| Mn (µg/L) | −11,076.50 | 6.2 | 6.2 |
| Ca (µg/L) | −189,357 | 190 | 190 |
| K (µg/L) | −1,854,423 | 2206 | 2162 |
| P (µg/L) | −49,117.0 | 138.8 | 134.5 |
| Composite desirability | 1.00 | 0.68 | 0.64 |
| Optimal value | | | |
| Trt (s) | 488 | 488 | 93.01 |
| E (kJ) | 13.2 | 13.2 | 2.54 |
| EFS (kV) | 0 | 30.99 | 31 |

Trt: Treatment time (s). E: Energy (kJ). EFS: Electric field strength (kV).

Color properties and especially the color density of the PEF treated (20 kV/cm, 0.5 Hz, 10 µs exponential decay pulses) red wine were reported very much similar to that of the untreated ones, ranging from 2.2 to 2.6. However, a slight decrease in color intensity in PEF-treated SO$_2$-free red wine samples was observed after over a week of storage [16].

Compared to the control, anthocyanin concentrations were superior (by up to 5% for *O. oeni* 9304 wines) in PEF-treated wines. Tannin concentrations after PEF treatment

showed a slight increase (up to 4 and 8% for *O. oeni* 9304 and *P. parvulus*, respectively), whereas the color intensity of the wines decreased by up to 2% for *B. bruxellensis* wines or increased by up to 5% for *O. oeni* 0608 wines [16]. TPSC of the PEF-treated samples was similar to those treated with $SO_2$ and untreated for six months of storage [18]. Similar to our results, the TPSC of the red wine samples was not significantly changed by PEF treatment (31 kV/cm, 30 °C, 3 µs square bipolar pulses, 40 mL/min) [10]. However, a 5% reduction in TPSC of the PEF-treated (20 kV/cm, 0.5 Hz, 10 µs exponential decay pulses) red wine samples was reported after eight days of storage [16]. Moreover, PEF application at different intensities at the end of the maceration step resulted in the improvement of color intensity, antioxidant capacity, and total phenolic index due to improved mass transfer in the winemaking process [12].

The color of the red wine is influenced by the anthocyanins as the anthocyanins are the main pigments responsible for the color. The stability of anthocyanins is affected by various parameters including pH, the presence of light, enzymes, type of metals, oxygen, storage temperature as well as processing conditions [19–21]. As changes in pH and metal ion concentration as well as processing temperature were not significantly changed by PEF treatment, TMAC was not adversely affected in the present study. The anthocyanin concentration of PEF-treated red wine was similar or superior to that of the control samples (by up to 5% for *O. oeni* 9304 wines). The tannin concentration was slightly higher in PEF-treated samples (up to 4 and 8% for *O. oeni* 9304 and *P. parvulus*, respectively) than that of the control samples. Compared to control samples, a slight decrease (by up to 2% for *B. bruxellensis* wines) or an increase (by up to 5% for *O. oeni* 0608 wines) was observed in the color intensity of the PEF-treated red wine samples. The total phenolic index of the PEF-treated wines either showed an increase (by up to 5% for *B. bruxellensis* wines) or a decrease (by up to 6% for *O. oeni* 9304 wines) in comparison with the control wines [16].

Spoilage microorganisms are one of the serious problems in the wine industry [22], and the addition of $SO_2$ is a common practice to decrease the risk of microbial spoilage during the winemaking process even though the sensitivity of microorganisms to $SO_2$ varies [23]. Moreover, due to the adverse health effects of $SO_2$, its reduction is recommended by WHO [24]. Efforts to eliminate $SO_2$ usage such as filtration [25] are not totally effective in addition to negatively affecting the physicochemical and sensory properties of the wine [26]; however, PEF provides inactivation of wine spoilage bacteria, wild yeasts as well as fermentative yeast, *S. cerevisiae*, for pasteurization purposes and reduces the amount of $SO_2$, which adversely affects the quality of wine [27].

Inactivation of wine spoilage yeasts revealed different results depending on the treatment parameters. Reported 0.8 log reduction in the mean initial number of *B. bruxellensis* was insufficient in PEF treated (250 Hz, 32 kV/cm, 30 pulses with 51.2 µs) red wine samples to prevent spoilage [18]; however, it was reported that passing red wine samples multiple times throughout the PEF treatment chambers to increase the treatment time (50 kV/cm, 100 Hz, 78 µs treatment time) would provide more than 6 log reduction on *B. bruxellensis* [18]. In fact, PEF treatment of red wine by 20 kV/cm, 0.5 Hz with 320 kJ/L energy was sufficient to obtain >4.8 log reduction on *B. bruxellensis* and a satisfactory inactivation on *P. parvulus*, *O. oeni* 9304, and *O. oeni* 0608 [16]. PEF treatment of red wine by 31 kV/cm at 30 °C with 500 Hz frequency provided more than 5 log reductions on *H. anemola*, *S. cerevisiae*, and *C. lipolytica*, in addition to 3.7 log reduction on *E. coli* O157:H7 and 3.46 log reduction on *L. delbrueckii* ssp. *bulgaricus* cultures, respectively [10]. In fact, 5.2 and 5.8 log reductions on *D. bruxellensis* and *D. anomala* were possible after red wine was exposed to 31 kV/cm, 1 Hz, and 100 pulses with processing temperatures of <30 °C [9].

Compared to control samples PEF treatment of red wine by 31 kV/cm revealed no significant difference in selected sensory properties [10]. No significant difference in the sensory properties of the control and the PEF-treated (250 Hz, 32 kV/cm, 30 pulses with 51.2 µs treatment time) red wine samples was also reported by the sensory panel even after one-year storage [18]. The fact that PEF-treated samples were evaluated as sweeter in the present study can be related to the increase in glucose-fructose levels after PEF

treatment. PEF-treated red wine samples inoculated with *B. bruxellensis* presented an increase in glucose-fructose levels up to 50% and residual reducing sugar concentrations up to 56% [16].

Both stability and organoleptic properties of wine are affected by various parameters including total and volatile acidity, pH, residual reducing sugar, glucose-fructose concentration, and organic acids and their concentrations. The level of acidity has a direct effect on sensory perception, the stability of tartrates and proteins, and the color of wine. pH is the result of the acids after disassociation and release as hydrogen ions, whereas total acidity is the sum of all organic acids and their salts. As the pH of the wines was slightly modified by PEF treatments; the total acidity decreased compared to the untreated wine except for *O. oeni* 0608 inoculated wines. Changes in the concentration of both lactic acid and tartaric acid have been reported on PEF-treated samples correlated with the sensory properties of a lively, sharp, and tingling sensation. Variations in the acidity, thus, have a further effect on organoleptic consequences [16].

## 4. Conclusions

It is revealed by this study that PEF treatment is a viable option to process red wine samples without adversely affecting the quality and sensory properties with the inactivation of spoilage and pathogenic microorganisms. Results obtained in this study were generally in parallel to the previous studies regarding the preservation of quality and sensory properties, but it should be kept in mind that the effect of PEF on red wine samples is highly dependent on the applied processing parameters as well as a variety of grapes and must composition. Generally, color enhancement associated with the increase in anthocyanin content correlated with the positive impact on health was observed in all studies related to increased electroporation on the plant cell. Even though microbial inactivation studies are very promising, further studies with process optimization are required to obtain >5 log reduction in microorganisms of interest in addition to shelf-life studies to determine the long-term effect of PEF on wine quality.

**Supplementary Materials:** The following supporting information can be downloaded at: https://www.mdpi.com/article/10.3390/beverages8040078/s1, Table S1: ANOVA results and estimated regression coefficients for the physical properties of red wine samples treated by pulsed electric; Table S2: ANOVA results and estimated regression coefficients for the total antioxidant capacity (TAC), total phenolic substance content (TPSC), and total monomeric anthocyanin content (TMAC) of the red wine samples treated by pulsed electric fields; Table S3: ANOVA results and estimated regression coefficients for the microbial inactivation of red wine samples treated by pulsed electric fields.

**Funding:** The research was funded by TUBITAK (project no: 104 O 585) and Bolu Abant Izzet Baysal University Research Fund (project no: 2009.09.01.307).

**Data Availability Statement:** Datasets generated and/or analyzed during the current study are available from the corresponding author upon reasonable request.

**Acknowledgments:** Red wine samples were kindly provided by Dimes Gida San ve Tic A.S. (Tokat, Turkey).

**Conflicts of Interest:** The author declares no conflict of interest.

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
