# Peer review of "Pulsed Electric Field Processing of Red Wine: Effect on Wine Quality and Microbial Inactivation"

_beverages, doi:10.3390/beverages8040078_

Round 1

Reviewer 1 Report

This paper covers a novel approach using pulsed electric field treatments to inactivate yeasts while enhance the quality of wine. The topic is very interesting, the paper is well written, and the results are useful, so the paper has merit to publish in this journal after minor revision.

Suggestions for revision:

Line 83: transferring them from WHERE to TSB?

Line 101-104: what are the inlet temperature and outlet temperature of wine sample?

Line 121: replace “conducted” with “determined”

Line 125: replace “.” with “,”

Line 124-131: list the temperature(s) for these measurements as their values are related to the temperatures.

Line 209: Please explain/discuss why PEF processing caused significant increase of pH and conductivity and any mechanisms behind them;  references are needed.

Author Response

Reviewer 1

This paper covers a novel approach using pulsed electric field treatments to inactivate yeasts while enhance the quality of wine. The topic is very interesting, the paper is well written, and the results are useful, so the paper has merit to publish in this journal after minor revision.

Suggestions for revision:

Comment 1. Line 83: transferring them from WHERE to TSB?

Response to comment 1: The sentence was corrected as;

“The yeasts were activated by transferring them from tryptic soy agar (TSA, Fluka, Munich, Germany) slants into tryptic soy broth (TSB) (Fluka) and following incubation at 22 ± 2 °C for 12 h.”

Comment 2. Line 101-104: What are the inlet temperature and outlet temperature of wine sample?

Response to comment 2:  The sentence was reorganized as follows;

“K-type dual channel digital thermocouples (Fisher Scientific, Pittsburgh, PA) was placed at the inlet and the outlet of each pair of treatment chambers to monitor the pre- and post-treatment temperatures (T2–T1, T4–T3 and T6–T5) of 14-12, 14-13, 14-13 ºC, respectively.”

Comment 3. Line 121: replace “conducted” with “determined”

Response to comment 3: It was changed to determined.

Comment 4. Line 125: replace “.” with “,”

Response to comment 4: It was replaced with “,”

Comment 5. Line 124-131: List the temperature(s) for these measurements as their values are related to the temperatures.

Response to comment 5: All measurements were conducted at room temperature (22 ± 2 ºC) and this information was added to the text.

Comment 6. Line 209: Please explain/discuss why PEF processing caused significant increase of pH and conductivity and any mechanisms behind them; references are needed.

Response to comment 6: The following text was added to the manuscript.

“Increase in pH can be observed as a consequence of several reactions including metal ion migration from the electrodes, growth of the yeasts as they have ability to degrade organic acids [18], and the leakage of intercellular components from cells to liquid phase as a result of increased membrane permeability. In parallel to increase in pH, conductivity of the samples was also significantly increased by applied magnitude of energy.”

Reviewer 2 Report

Very long paragraphs and   difficulties in understanding at 

 Line 32 to line 42 

 Line 47  to line 59 

 It has  to change at the following  points:   

Line 54  

 in  ‘as well as’

Line 61

‘extraction efficacy of bioactive’ in  ‘efficacy in extraction of bioactive’

Line 67 

‘by PEF to reduce SO2 use and determine’ in ‘by  means of PEF to reduce SO2 use and to determine’

Line 121

‘pH of the samples was conducted with 10 mL of the samples’ in ‘pH   was  determined on  10 mL    samples’

Line 125

Eliminate fullstop  in ‘acid (g/L). whereas’

Line 128

‘was measured’ in  ‘were  measured’

Line 133

‘with the samples diluted‘  in  ‘on   samples diluted’

Line 164

‘plates, for H. anomala’ in ‘plates. Dilutions  for H. anomala’

Line 166

‘(Fluka), for L. delbrueckii ssp.’ in  ‘(Fluka) and dilutions for  L. delbrueckii ssp.’

 Line 180

‘breath to receive the whole’ in  breath for  receiving  the whole’

Line 186-187

‘They have drunken water between the samples, consume 186 couple bits of unsalted cracker and drink water again before the 187 evaluation of the next sample.’  in ‘Among the samples evaluation    they  drunk water,  consumed  couple bits of unsalted cracker and drunk water again.

Line 190-192

 Data were analyzed with Tukey’s multiple comparison tests 191 followed by one-way analysis of variance (ANOVA) to determine the 192 mean significant differences in terms of quality properties (pH, TA, 193 conductivity’  in  ‘Data were analyzed by one-way analysis of variance (ANOVA) for the evaluation  of  the  significance (F  test) and  Tukey’s multiple comparison test was utilized  to determine the   means significant differences   in quality properties (pH, TA, 193 conductivity, … ‘

 These points are  not clear:

Line 63-64

‘but PEF treatment of red wine samples with  determination of changes in physicochemical and sensory properties 65 in addition to microbial inactivation’

 Line ? at pag. 18

The anthocyanin concentration of PEF treated red wine were similar or superior to that of the control samples (by up to 5% for O. oeni 9304 wines).

Do  you mean ‘was’ or ‘were’?  May it be ‘concentrations’ and  ‘wines’ ?

 Line ? at pag.19

‘PEF treatment of red wine by 31 kV/cm at 30 °C with 500 Hz frequency provided more than 5 log’

Why these size of characters?

 Line ? at pag.20

‘Results obtained in this study was generally was in parallel to the previous studies regarding preservation of quality and sensory properties’

Erase one ‘was’ of two. Then,  is it  ‘was’ or ‘were’ ?

 Author Response

Reviewer 2

Comment 1: Very long paragraphs and difficulties in understanding at Line 32 to line 42 and Line 47 to line 59 

Response to comment 1: Sensory properties of particle status, sour taste and after taste were significantly decreased; whereas the other measured properties were significantly increased by 13.2 kJ PEF treatment (p < 0.05). Joint optimization studies for the most optimal processing parameters for measured properties were 488 sec, 0.13 kJ, and 0.22 kV; 488 sec, 13.2 kJ, and 31 kV; 348 sec, 9.39 kJ, and 31 kV/cm; and 488 sec, 13.2 kJ, and 0 kV EFS, with 0.79, 0.69, 1.00 and 0.72 composite desirabilities, respectively.

Comment 2: It has to change at the following points: Line 54 in ‘as well as’

Response to comment 2: It was corrected as suggested.

Comment 3: Line 61 ‘extraction efficacy of bioactive’ in ‘efficacy in extraction of bioactive’

Response to comment 3: It was changed to ‘efficacy in extraction of bioactive’

Comment 4: Line 67 ‘by PEF to reduce SO2 use and determine’ in ‘by means of PEF to reduce SO2 use and to determine’

Response to comment 4: It was corrected as “by means of PEF to reduce SO2 use and to determine…”

Comment 5: Line 121 ‘pH of the samples was conducted with 10 mL of the samples’ in ‘pH was determined on 10 mL samples’

Response to comment 5: It is corrected as suggested.

Comment 6: Line 125 Eliminate fullstop in ‘acid (g/L). whereas’

Response to comment 6: It was deleted.

Comment 7: Line 128 ‘was measured’ in  ‘were  measured’

Response to comment 7: It was corrected as suggested.

Comment 8: Line 133 ‘with the samples diluted ‘in ‘on samples diluted’

Response to comment 8: It was corrected as suggested.

Comment 9: Line 164 ‘plates, for H. anomala’ in ‘plates. Dilutions for H. anomala

Response to comment 9: It was corrected as suggested.

Comment 10: Line 166 ‘(Fluka), for L. delbrueckii ssp.’ in  ‘(Fluka) and dilutions for  L. delbrueckii ssp.’

Response to comment 10: It was corrected as suggested.

Comment 11: Line 180 ‘breath to receive the whole’ in ‘breath for receiving the whole’

Response to comment 11: It was corrected as suggested.

Comment 12: Line 186-187 ‘They have drunken water between the samples, consume couple bits of unsalted cracker and drink water again before the evaluation of the next sample.’  in ‘Among the samples evaluation they drunk water, consumed couple bits of unsalted cracker and drunk water again.

Response to comment 11: The sentence was corrected as suggested.

Comment 12: Line 190-192  ‘Data were analyzed with Tukey’s multiple comparison tests 191 followed by one-way analysis of variance (ANOVA) to determine the mean significant differences in terms of quality properties (pH, TA, conductivity’  in  ‘Data were analyzed by one-way analysis of variance (ANOVA) for the evaluation  of  the  significance (F  test) and  Tukey’s multiple comparison test was utilized  to determine the   means significant differences   in quality properties (pH, TA, 193 conductivity, … ‘

Response to comment 12: The sentence was corrected as suggested.

Comment 13: These points are not clear: Line 63-64

‘but PEF treatment of red wine samples with determination of changes in physicochemical and sensory properties in addition to microbial inactivation’

Response to comment 13: The sentence was completed as “… but PEF treatment of red wine samples with determination of changes in physicochemical and sensory properties in addition to microbial inactivation are very limited.”

Comment 14: Line ? at page 18

The anthocyanin concentration of PEF treated red wine were similar or superior to that of the control samples (by up to 5% for O. oeni 9304 wines).

Do  you mean ‘was’ or ‘were’?  May it be ‘concentrations’ and  ‘wines’ ?

Response to comment 14: It was corrected as “The anthocyanin concentration of PEF treated red wine was…”

Comment 15: Line ? at page19

‘PEF treatment of red wine by 31 kV/cm at 30 °C with 500 Hz frequency provided more than 5 log’ Why these size of characters?

Response to comment 15: They were corrected.

Comment 16: Line ? at page 20

‘Results obtained in this study was generally was in parallel to the previous studies regarding preservation of quality and sensory properties’

Erase one ‘was’ of two. Then, is it  ‘was’ or ‘were’ ?

Response to comment 16: second “was” was deleted and corrected as “were”.

Reviewer 3 Report

In the article, Gulsun Akdemir Evrendilek studies the effectiveness of Pulsed Electric Field (PEF) to inactivate the microorganisms that are involved in the production of wine. The motivations for this tool are clearly stated, the wine industry is interested in reducing the amount of chemical additives in wine that are used to eliminate microbes. The researchers wanted to optimize the inactivation of the microbes while trying to minimize the tradeoff of wine quality.

There have been studies in the past that show the power of PEF to inactivate microbes and extract phenolics and anthocyanins, but this paper intentionally tries to optimize the PEF treatment for red wine without sacrificing the properties of wine that interest consumers such as mouthfeel and other sensory properties. The author did a great job of citing the past work using PEF in wine processing. They also did a great job explaining how their work adds to the knowledge but is asking a specific, novel question. They also nicely state the future directions that are possible by stating specific aims for increased optimization.

Since this article is somewhat of a straightforward report of the effect of PEF on red wine processing it is not surprising that there are multiple pages of charts. These charts detail the effect of different PEF treatments on wine microbiota to test for effectiveness of processing but also show statistics on what is significant and not significant. I appreciate that the author prepared a careful and detailed note about these data. In the comments I suggest ways to improve the flow of the report and allow it to read more like an article, as of now the figures just feel like raw data.

            Specific comments:

1.     I did not enjoy the way that the article was structured. I feel as though the figures really interrupt the flow of the article because they do not provide any information that you cannot find directly in the text. They are bulky charts and their placement felt random. My suggestion is to present the data in a more engaging way in the figures, maybe an illustration or a graph instead of merely a chart restating the info that is already in the text. If you want to include the chart the way it is now, maybe reduce the amount of “reporting” in the text, focusing more on the significant results. The way the data is presented really jams up the report which is actually straightforward.

2.     This is a more minor aesthetic issue but was just as distracting to me as the reader. Some charts get a full page to itself and are presented in landscape mode, while others are integrated in the actual text. You should pick one, consistent way of presenting the data.

3.     It would be effective to move all statistical analysis such as the ANOVA table to supplementary figures. Again, even though it is important to the results of the study the statistical information doesn’t help the flow of the article or actually report any information that the reader is curious about.

Author Response

In the article, Gulsun Akdemir Evrendilek studies the effectiveness of Pulsed Electric Field (PEF) to inactivate the microorganisms that are involved in the production of wine. The motivations for this tool are clearly stated, the wine industry is interested in reducing the amount of chemical additives in wine that are used to eliminate microbes. The researchers wanted to optimize the inactivation of the microbes while trying to minimize the tradeoff of wine quality.

There have been studies in the past that show the power of PEF to inactivate microbes and extract phenolics and anthocyanins, but this paper intentionally tries to optimize the PEF treatment for red wine without sacrificing the properties of wine that interest consumers such as mouthfeel and other sensory properties. The author did a great job of citing the past work using PEF in wine processing. They also did a great job explaining how their work adds to the knowledge but is asking a specific, novel question. They also nicely state the future directions that are possible by stating specific aims for increased optimization.

Since this article is somewhat of a straightforward report of the effect of PEF on red wine processing it is not surprising that there are multiple pages of charts. These charts detail the effect of different PEF treatments on wine microbiota to test for effectiveness of processing but also show statistics on what is significant and not significant. I appreciate that the author prepared a careful and detailed note about these data. In the comments I suggest ways to improve the flow of the report and allow it to read more like an article, as of now the figures just feel like raw data.

Specific comments:

Comment 1: I did not enjoy the way that the article was structured. I feel as though the figures really interrupt the flow of the article because they do not provide any information that you cannot find directly in the text. They are bulky charts and their placement felt random. My suggestion is to present the data in a more engaging way in the figures, maybe an illustration or a graph instead of merely a chart restating the info that is already in the text. If you want to include the chart the way it is now, maybe reduce the amount of “reporting” in the text, focusing more on the significant results. The way the data is presented really jams up the report which is actually straightforward.

Response to comment 1: The Beverage Journal format requires to insert tables and figures in the text; thus, text is interrupted by the tables. It is possible to convert 2-3 tables into figures but the figures will be too crowded and hard to understand. If it is possible, tables 6-8 can be given as supplementary data.

Comment 2: This is a more minor aesthetic issue but was just as distracting to me as the reader. Some charts get a full page to itself and are presented in landscape mode, while others are integrated in the actual text. You should pick one, consistent way of presenting the data.

Response to comment 2: All tables collected before discussion section and arranged as landscape format so that the manuscript will have more aesthetic look.

Comment 3. It would be effective to move all statistical analysis such as the ANOVA table to supplementary figures. Again, even though it is important to the results of the study the statistical information doesn’t help the flow of the article or actually report any information that the reader is curious about.

Response to comment 3: Tables 6-8 can be given as supplementary data.
